# Rhinacanthin-C but Not -D Extracted from *Rhinacanthus nasutus* (L.) Kurz Offers Neuroprotection via ERK, CHOP, and LC3B Pathways

**DOI:** 10.3390/ph15050627

**Published:** 2022-05-20

**Authors:** Varaporn Rakkhittawattana, Pharkphoom Panichayupakaranant, Mani I. Prasanth, James M. Brimson, Tewin Tencomnao

**Affiliations:** 1Ph.D. Program in Clinical Biochemistry and Molecular Medicine, Department of Clinical Chemistry, Faculty of Allied Health Sciences, Chulalongkorn University, Bangkok 10330, Thailand; ovr_1129@hotmail.com; 2Natural Products for Neuroprotection and Anti-Ageing Research Unit, Chulalongkorn University, Bangkok 10330, Thailand; prasanth.m.iyer@gmail.com; 3Department of Pharmacognosy and Pharmaceutical Botany, Faculty of Pharmaceutical Sciences, Prince of Songkla University, Songkhla 90112, Thailand; pharkphoom.p@psu.ac.th; 4Department of Clinical Chemistry, Faculty of Allied Health Sciences, Chulalongkorn University, Bangkok 10330, Thailand

**Keywords:** autophagy, Parkinson’s, glutamate toxicity, neurodegeneration, neuroprotection

## Abstract

Neurodegenerative diseases present an increasing problem as the world’s population ages; thus, the discovery of new drugs that prevent diseases such as Alzheimer’s, and Parkinson’s diseases are vital. In this study, Rhinacanthin-C and -D were isolated from *Rhinacanthus nasustus*, using ethyl acetate, followed by chromatography to isolate Rhinacanthin-C and -D. Both compounds were confirmed using NMR and ultra-performance-LCMS. Using glutamate toxicity in HT-22 cells, we measured cell viability and apoptosis, ROS build-up, and investigated signaling pathways. We show that Rhinacanthin-C and 2-hydroxy-1,4-naphthoquinone have neuroprotective effects against glutamate-induced apoptosis in HT-22 cells. Furthermore, we see that Rhinacanthin-C resulted in autophagy inhibition and increased ER stress. In contrast, low concentrations of Rhinacanthin-C and 2-hydroxy-1,4-naphthoquinone prevented ER stress and CHOP expression. All concentrations of Rhinacanthin-C prevented ROS production and ERK1/2 phosphorylation. We conclude that, while autophagy is present in HT-22 cells subjected to glutamate toxicity, its inhibition is not necessary for cryoprotection.

## 1. Introduction

Diseases caused by neurodegeneration, including Alzheimer’s disease (AD), Parkinson’s disease (PD), multiple sclerosis (MS), and amyotrophic lateral sclerosis (ALS), have become significant health problems as the world’s population has aged. Currently, the number of people living with AD and PD is estimated to be 44 and 7 million, respectively, worldwide [1,2]. As life expectancy increases worldwide, the cases are expected to increase to 123 million and 20 million, respectively, by 2050 [3]. There are currently limited treatment options available for neurodegenerative disease, so the search for more effective drugs and therapies is vital [4]. The development of new drug compounds is costly and time-consuming. Prominent medicinal chemists suggest that producing a novel chemical entity in drug development requires seven or more synthetic steps [5,6]; thus, it is essential to take advantage of natural products to identify new drug-like compounds. As many natural products and herbal extracts have been utilized in traditional medicine for thousands of years, and there are currently at least 119 compounds in medical use that have been derived from plants [7]; furthermore, it is estimated that up to four billion people in the developing world rely on medicinal plants for their health care [8].

Oxidative stress and glutamate toxicity are thought to play a role in several neurodegenerative diseases, including MS, ALS, PD, and AD [9]. Several forms of cell death may be involved in neurodegenerative disease, including apoptosis (from the Greek, falling-off and referring to “programmed cell death”), autophagy (literally meaning “self-eating”), and necrosis (meaning “dead”) along with others [10]. The use of HT-22 cells subjected to glutamate toxicity is a commonly used model to investigate the neuroprotective effects of drugs against oxidative stress, as glutamate inhibits the cystine import, preventing the production of glutathione, a cellular antioxidant. The mechanism of cell death in this model has been shown to primarily include apoptosis, necrosis [11], and autophagic cell death [12].

Various herbal extracts are protective against glutamate toxicity. The medicinal herb, *Rhinacanthus nasutus* (L.) Kurz (Acanthaceae) (Rn) is widely seen in Thailand and Southeast Asia; it has many English common names, including Dainty spurs, Snake Jasmine, and White Crane flower. Several natural products have been isolated from Rn, with some of the most abundant being the rhinacanthins [13] (Figure 1), including Rhinacanthin-C (Rn-C), Rhinacanthin-D (Rn-D) [14], and Rhinacanthin-N (Rn-N) [15]. Previous studies from our group have shown that crude extracts of leaves and roots of Rn have neuroprotective effects against glutamate, hypoxia, and β-amyloid toxicity in HT-22 cells [16,17]. Furthermore, recent research has shown that Rn-C attenuates neuroinflammation triggered by Lipopolysaccharide (LPS), amyloid-β, and interferon-γ in BV-2 and primary rat glial cells and protects primary rat neurons from damages brought on by β-amyloid toxicity [18]. Rn-C has also been shown to have antioxidant activities and can prevent interleukin-1b (IL-1b), interleukin-6 (IL-6), and tumor necrosis factor-alpha (TNF-α) mRNA expression in animals subjected to subarachnoid hemorrhage [19].

This study aims to identify whether Rn-C, Rn-D, and 2-hydroxy-1,4-naphthoquinone (NpQ) are responsible for the neuroprotective effects exhibited by the crude extracts of Rn. 

## 2. Materials and Methods

### 2.1. Chemicals and Reagents

The antibodies for western blotting, including C/EBP homologous proteins (CHOP), extracellular signal-related kinases ½ (ERK1/2), phosphorylated extracellular signal-related kinases ½ (pERK1/2), light chain-3 beta (LC3B), β-actin, and the horseradish peroxidase (HRP) conjugated secondary antibody were all purchased from Cell Signaling Technology via Theera Trading. Co., Ltd. (Bangkok, Thailand). Antibodies for confocal microscopy were obtained from Sigma Aldrich (St. Louis, MI, USA). (LC3B anti-rabbit-FITC), Cell Signaling Technology (Danvers, MA, USA) (Anti-mouse-Alexa Fluor 555), or Merck (p62 (sequestosome-1)). 2-hydroxy-1,4-naphthoquinone was bought from Sigma-Aldrich (St. Louis, MI, USA). 3-(4,5-Ethylthiazol-2-yl)-2,5-diphenyltetrazolium Bromide (MTT) was purchased from MERCK (Kenilworth, NJ, USA). Hoechst 33342 was acquired from Sigma Aldrich (St. Louis, MO, USA).

### 2.2. Plant Material and Rhinacanthin Extraction

Rn was collected from the Jana district of Songkhla Province in Thailand. The samples were authenticated at Prince Songkla University herbarium, Thailand, and given specimen voucher no. 001-18-14. The leaves were washed and dried, then ground into a fine powder. Ethyl acetate was chosen as the solvent for extraction as previous studies have shown a high yield of rhinacanthin compounds [15]. The powdered Rn leaves were refluxed in ethyl acetate (1 Kg leaf:5 L ethyl acetate) for 1 h, three times. The extract was then dried, and five grams of the crude extract was dissolved in 200 mL of methanol and loaded onto an anion exchange resin column (Amberlite^®^ IRA-67). The Column was washed with methanol before the rhinacanthin-rich fraction was eluted using 10% acetic acid in methanol; this fraction was concentrated using a rotary evaporator for further purification using column chromatography.

Rn-C and Rn-D were isolated using Liquid chromatography with silica gel as the solid phase, and hexane: ethyl acetate (18:1) was used as the mobile phase. Fractions of 30 mL were collected and analyzed with thin layer chromatography (TLC). Matching fractions were pooled and then separated with a Sephadex LH20 column using methanol as the mobile phase. Fractions were again analyzed with TLC and re-fractioned until each spot remained for Rn-C and Rn-D. The individual fractions for each Rh-C or Rh-D were collected separately and dried using a rotary evaporator.

### 2.3. Rhinacanthin Purity Analysis—Recycling Preparative-HPLC

The isolated rhinacanthins were analyzed using the recycling preparative high-performance liquid chromatography (HPLC) system (JAI LC-91NEXT, (Japan Analytical Industry, Tokyo, Japan)). The isolated rhinacanthins were reconstituted in methanol, filtered through a 0.2 µm filter, loaded onto a C18 column (JAIGEL-ODS 21.5 mm × 300 mm), and subjected to recycling preparative HPLC using 1% acetic acid in methanol as the mobile phase at a flow rate of 5 mL/min.

### 2.4. Nuclear Magnetic Resonance Analysis

The isolated rhinacanthin structures were analyzed using nuclear magnetic resonance (NMR). One dimensional ^1^H-NMR analysis was performed using NMR (Bruker Advance 500) at 400 MHz, with the rhinacanthins dissolved in Deuterated chloroform (CDCl_3_). The ^1^H-NMR spectrum profiles were analyzed with ACD/NMR processor Academic Edition version 12.01 (Advanced Chemistry Development, Toronto, ON, Canada).

The NMR analysis for RnC and RnD can be found in Appendix A, respectively. ^1^H-NMR analysis of Rn-C resulted in δ (ppm): 

1.03 (s, 6 H), 1.55–1.61 (m, 6 H), 1.80 (d, J = 1.01 Hz, 3 H), 2.00–2.06 (m, 2 H), 2.14–2.22 (m, 2 H), 2.71 (s, 2 H), 3.92 (s, 2 H), 5.22 (dd, J = 6.57, 1.26 Hz, 1 H), 6.70 (td, J = 7.26, 1.39 Hz, 1 H), 7.39 (s, 1 H) 7.63–7.81 (m, 2 H) 8.12 (dd, J = 7.71,1.14 Hz, 1 H), 8.10 (dd, J = 7.58, 1.26 Hz, 1 H).

The pure Rn-D showed the signal of ^1^H-NMR at δ (ppm):

1.09 (s, 6 H) 2.17 (s, 1 H), 2.76 (s, 2 H), 4.06 (s, 2 H), 6.00 (s, 2 H), 6.74 (d, J = 8.34 Hz, 1 H), 7.41 (d, J = 1.77 Hz, 1 H), 7.50—7.74 (m, 3 H), 8.04—8.09 (m, 2 H).

### 2.5. Ultrahigh-Performance Liquid Chromatography-Mass Spectrometry 

Ultrahigh-performance Liquid Chromatography-mass spectrometry (UHPLC-MS) was done using a Thermo Exactive Orbitrap mass spectrometer linked with an Accela 600 ultra-performance liquid chromatography (UPLC) pump and an Accela autosampler (ThermoFisher Scientific, Waltham, MA, USA). Chromatographic separation was performed on a Kromasil C18 250 × 4.6 mm × 5 u column at 350 °C with a 0.4 mL/min flow rate and injection volume of 20 µL. A Peak Scientific NM32LA nitrogen generator (Peak Scientific Instruments, Inchinnan, UK) was used to produce Nitrogen, and a heated electrospray (HESI) with the transfer line temperature set to 350 °C was used to induce ionization along with a spray voltage of 4 kV. Positive mode analysis was done in a mobile gradient phase using binary solvents. Mobile phases A and B had 0.1% formic acid, with H_2_O and Methanol respectively. The mobile phase B was varied from 0 to 95% from 0 to 50 min, 95% B from 50–55 min, and 55.1–60 min to initial conditions. Negative ionization mode used a similar program with water as mobile phase A and acetonitrile as mobile phase B.

High-resolution mass spectrometry (HRMS) detected the molecular formula of compounds by comparing theoretical and observed mass.

### 2.6. Cell Culture 

The HT-22 cell line was a gratefully received donation from Professor David Schubert at the Salk Institute (San Diego, CA, USA). Dulbecco’s modified eagle medium (DMEM) was used to maintain the cultured cells, supplemented with 10% heat-inactivated fetal bovine serum (FBS), penicillin (50 I.U./mL), and streptomycin (50 μg/mL). Cells were sub-cultured every seven days, with the media being replaced every two to three days. All cell lines were maintained in a humidified air environment supplemented with 5% CO_2_ at 37 °C.

### 2.7. Cell Viability Assay

Cells were plated at a density of 15,000 cells/cm^2^ (≈5000 cells per well) and allowed to adhere overnight in 96-well plates. The following day, glutamate (5 mM) with or without varying concentrations of Rn-C, Rn-D, or NpQ (0.3125 to 10 µM) was added. Control cells were treated with an equal volume of solvent in phosphate-buffered saline (PBS). The cells were then incubated for 24 h. The Media was replaced (with the supernatant kept for analysis in the LDH assay) before the addition of 5 mg/mL (12 mM) MTT solution (final concentration 0.5 mg/mL (1.2 mM). The cells were then incubated for 4 h at 37 °C in a humidified air atmosphere supplemented with 5% CO_2_ to allow the formazan salt to develop. The formazan was then solubilized in Dimethyl sulfoxide (DMSO), and the absorbance was read at 550 nm using a microplate reader. LDH release was measured using the non-radioactive cytotoxicity assay (Promega, Madison, WI, USA) per the manufacturer’s instructions.

### 2.8. Annexin V/Propidium Iodide Staining

HT-22 were seeded on 6-well plates at a density of 5200 cells/cm^2^ (≈50,000 cells/well). Cells were treated with glutamate or co-treated with Rh-C and glutamate at different concentrations for 24 h. Then cells were stained with Annexin A5 Apoptosis Detection Kit (Biolegend, SanDiego, CA, USA) with a slightly modified staining protocol. The media was removed, and cells were trypsinized and washed twice with ice-cold PBS. Then cells were spun at 4000 rpm for five min at 4 °C, resuspended, and stained with 2.5 µL of Annexin V and 5 µL of propidium iodide solution. The mixture was vortexed and incubated for 15 min at room temperature in the dark. Subsequently, 400 µL of Annexin V binding buffer was added and apoptosis was analyzed with flow cytometry (FACSCalibur, BD Biosciences, San Jose, CA, USA).

### 2.9. Reactive Oxygen Species Analysis 

Glutamate has been shown in previous studies to induce the production of reactive oxygen species (ROS) in HT-22 cells [16,20,21]. ROS was measured using H_2_DCFDA following the previously described protocol [17]. Briefly, HT-22 cells were seeded at 30,000 cells/cm^2^ (≈10,000 cells per well) and allowed to adhere overnight in nonfluorescent 96-well plates before treatment with 5mM glutamate and/or Rn-C. After 18 h of treatment, cells were carefully washed with Hank’s buffered saline solution (HBSS) and loaded with 10 µM Carboxy-H_2_DCFDA (Invitrogen, Waltham, MA, USA) in prewarmed HBSS. Cells were subsequently incubated at 37 °C with a humidified 5% CO_2_ atmosphere for 30 min before being washed in warm HBSS three times and fluorescence measured (ex.494–em.521) using a plate reader. Data were normalized with control cells.

### 2.10. Western Blotting

HT-22 cells at a density of 10,000 cells/cm^2^ in six-well plates (≈100,000 cells per well) were allowed to adhere overnight before treatment with glutamate and/or Rn-C, Rn-D, or NpQ for 18 h. The cells were washed thoroughly (three times) in chilled PBS. The total cellular protein was isolated using radioimmunoprecipitation assay buffer (RIPA) lysis buffer (Merck Millipore, Burlington, MA, USA) with protease inhibitor cocktail (Roche, Basel, Switzerland) with added phosphatase inhibitor cocktail. Protein concentrations were analyzed using the Bradford assay (Biorad, Hercules, CA, USA). The protein samples were then separated (in equal concentrations) using sodium dodecyl sulfate-polyacrylamide gel electrophoresis (SDS-PAGE) (10% SDS gel) at 70V for 30 min and 120 V for the next 60 min. The protein was then transferred to 0.45 µm polyvinylidene difluoride (PVDF) membranes (GE Healthcare, Chicago, IL, USA) at 150 mA for 90 min. Laboratory grade blocking reagent (Biorad, Hercules, CA, USA) was used to block the membranes for one hour before probing with antibodies. The membranes were probed with the following primary antibodies ERK1/2 (Thr202/Tyr204, 1:1000), LC3B (1:1500), ERK1/2 (1:1000), and β-actin (1:10,000) for 18 h at 4 °C with gentle rocking. Membranes were then washed 3 times for 10 min in Tris-buffered saline Tween 20 (TBS-T). The secondary antibody (anti-rabbit HRP-conjugated) was probed at room temperature for one hour and then washed three times in TBS-T. The membranes were visualized by treating the membranes with ECL-prime western blotting reagent (GE Healthcare, Chicago, IL, USA) and exposure to Hyperflim-ECL X-ray film (GE Healthcare, Chicago, IL, USA). Blots were then scanned and digitally analyzed with Image J software using β-actin as the loading control.

### 2.11. Immunofluorescence Colocalization of LC3B and p62 (Sequestosome-1)

HT-22 cells were fixed in 4% paraformaldehyde in PBS pH 7.4 for 10 min, then permeabilization using 0.5% Triton X-100 and blocking with 1% FBS for one hour with slight agitation. The slides were then washed three times with PBS and incubated overnight with the primary antibodies (LC3B and p62 (sequestersome-1). The slides were washed with PBS thrice and incubated with secondary antibodies (anti-rabbit FITC and anti-mouse-Alexa Fluor-555. Hoechst 33342 (0.2 mM) was added 5 min before mounting in SlowFade antifade reagent (Thermo Fisher Waltham, MA, USA). A confocal microscope (Carl Zeiss LSM700, Oberkochen, Germany) was used to obtain the fluorescence images at 100× magnification. The colocalization coefficient was calculated using Zen light 2009 software (Carl Zeiss, Oberkochen, Germany).

### 2.12. Statistical Analysis

All results were expressed as the mean ± SEM from at least three independent experiments. Wherever appropriate, statistical analysis was carried out using analysis of variance (ANOVA) followed by Dunnett’s post hoc test, with *p* values < 0.05 being considered significant. Statistical analysis and graphs were produced using Graph Pad Prizm (version 9 for Mac) (Graphpad Holdings San Diego, CA, USA).

## 3. Results

### 3.1. Identification, Yield, and Purity of Rn-C and Rn-D

Using 1 kg of dried Rn leaf powder yielded 48.7 mg of pure Rn-D and 1110 mg of Rn-C.

A comparison of NMR data to that of other studies indicated the structures of Rn-C and Rn-D to be as previously identified [14,22] (Appendix A). Mass spectrophotometry analysis of Rn-C (C_25_H_30_O_5_) resulted in 410.5 g/mol (Appendix A), and Rn-D resulted in 408 g/mol (Appendix A). Repeating HPLC analysis showed symmetrical peaks indicating pure isolation of Rn-C (Appendix A).

### 3.2. Neuroprotective Properties of Rn-C Rn-D and NpQ

The neuroprotective effects of Rn-C, Rn-D (isolated from Rn), and NpQ, the basic naphthoquinone that forms the core structure of Rn-C and Rn-D, were analyzed using glutamate toxicity in HT-22 cells. HT-22 cells subjected to 18 h of 5 mM glutamate showed a reduction in viability of over 80% (measured by MTT metabolism). Rn-C showed neuroprotective effects at concentrations as low as 312.5 nM, returning viability from 15.79 ± 1.55% to 50.31 ± 5.20% (Statistically significant with ANOVA Dunnett’s post hoc test *p*-value < 0.05). The highest effective concentration of Rn-C tested was 10 µM returning viability from 15.79 ± 1.55% to 63.4 ± 3.22% (Statistically significant with ANOVA Dunnett’s post hoc test *p*-value < 0.05) (Figure 2A). Furthermore, when assessing glutamate toxicity using the non-radioactive cytotoxicity assay, which measures LDH release (Figure 2B), all concentrations of Rn-C tested returned the levels of LDH release to that of the untreated control. 

On the other hand, Rn-D had a minimal protective effect against 5 mM glutamate, with only 10 µM Rn-D having any statistically significant increase in cell viability; however, the increase was modest as it improved cell viability from 13.11 ± 0.68% to 26.43 ± 3.15% (Statistically significant with ANOVA Dunnett’s post hoc test *p*-value < 0.05 *n* = 3) (Figure 2C). Furthermore, when the glutamate toxicity was measured with the LDH assay, Rn-D failed to have any protective effect (Figure 2D). NpQ was only effective at the highest concentration we tested in the MTT assay (Figure 2E), improving cell viability from 16.00 ± 2.14% to 88.00 ± 17.74% (Statistically significant increase with ANOVA Dunnett’s Post hoc test *p*-value < 0.05). There also appears to be a reduction in LDH release at all concentrations; however, these reductions were not statistically significant (Figure 2F).

Analysis of the cellular morphology of 5 mM glutamate treated HT-22 cells revealed a clear difference between control cells and glutamate treated cells (Figure 3A,B), with glutamate treated cells appearing small and round with no neuron-like definition and shape. Treatment with 0.1 µM Rn-C seemed to have a minimal protective effect (Figure 3C), with the cells appearing in the same morphology as the glutamate alone treated cells. Treatment with 1 and 10 µM Rn-C seems to defend the HT-22 cells against glutamate-induced morphology changes, with cells appearing much more like the control (Figure 3D,E). Rn-D and NpQ seemed not to change the morphology of HT-22 cells treated with glutamate (5 mM) except for at higher concentrations NpQ (10 µM), data not shown.

The staining of HT-22 cells with Annexin V and propidium iodide revealed that glutamate resulted in significant apoptotic cell death at concentrations between 5 and 10 mM (Figure 4A,B). Furthermore, Rn-C was able to prevent this apoptotic cell death in a dose-dependent manner, at doses ranging between 0.1 and 10 µM (Figure 4C,D). 

### 3.3. Reactive Oxygen Species Production

Previous studies with the crude extract of Rn have shown that Rn can reduce the level of ROS in stressed HT-22 cells [16,17,23]. Having seen the protective effects of Rn-C against glutamate-induced toxicity in HT-22 cells, we investigated whether Rn-C could have the same effect as Rn in mitigating ROS. Rn-C was able to dose-dependently reduce ROS resulting from 5 mM glutamate treatment in HT-22 cells with 5 µM Rn-C reducing ROS from 265 ± 19.61% to 176.60 ± 7.30% (Statistically significant reduction in ROS with ANOVA followed by Dunnett’s post hoc test *p* < 0.05) and 10 µM reducing ROS production from 265 ± 19.61% to 114.60 ± 10.44% (Statistically significant reduction in ROS with ANOVA followed by Dunnett’s post hoc test *p* < 0.05 *n* = 5) (Figure 5).

### 3.4. Cell Signaling in Response to Glutamate and Rn-C or NpQ

ERK1/2 activation is usually considered a pro-survival signal; however, prolonged activation has been shown to induce apoptosis [24,25,26]. Furthermore, several studies have shown that glutamate toxicity in HT-22 cells is mediated through ERK1/2 activation and that inhibiting ERK1/2 phosphorylation prevents cell death caused by glutamate toxicity [27,28,29]; therefore, we investigated whether Rn-C and NpQ affected ERK phosphorylation (Figure 6A). We found that while total ERK expression remained the same (Figure 6B), pERK expression was dramatically raised 4.42 ± 1.06-fold (ANOVA with Dunnett’s post hoc test *p* < 0.05 (*n* = 5)) (compared to control) 18 h after treatment with 5 mM glutamate. Rn-C (0.1 µM–10 µM) and NpQ (10 µM) prevented ERK activation with pERK levels remaining around the levels of the control (Figure 6C) (ANOVA with Dunnett’s post hoc test *p* < 0.05 (*n* = 5)). Moreover, NpQ appeared to reduce pERK levels to below that of the control with a 0.55 ± 0.20-fold change in pERK expression (ANOVA with Dunnett’s post hoc test *p* < 0.05 (*n* = 5)).

Previous studies have shown that glutamate induces endoplasmic reticulum (ER) stress in HT-22 cells, leading to cell death via caspase-dependent apoptosis [20,30,31]; therefore, we investigated the effects of Rn-C and NpQ on glutamate-induced CHOP expression (Figure 7). Glutamate treatment for 18 h resulted in a 5.96 ± 1.6-fold increase in CHOP expression. Co-treatment with glutamate (5 mM) and Rn-C 10 or 1 µM resulted in a 69.70 ± 18.10 and 57.15 ± 25.87-fold increase, respectively (Figure 7B); however, cotreatment with 0.1 µM Rn-C or 10 µM NpQ prevented the increase in CHOP expression (ANOVA with Dunnett’s post hoc test *p* < 0.05 (*n* = 3)) (Figure 7C).

In previous studies, glutamate activated autophagy in HT-22 cells [12,21], whereas CHOP expression had been shown to prevent autophagy [32,33,34,35] and ERK phosphorylation was regulated by autophagy proteins [36]; therefore, we investigated the effects of Rn-C and NpQ on autophagy induction in HT-22 cells subjected to 5 mM glutamate. LC3I conversion to LC3II is an indicator of autophagy induction, and thus, we measured the ratio of LC3II to LC3I via western blot (Figure 8A). We set the ratio to one for the control, and treatment with 5 mM glutamate resulted in a 2.33 ± 0.52-fold increase in LC3II/LC3I ratio (Figure 8B). Rn-C (1 and 10 µM) inhibited autophagy induction with the fold changes in LC3II/LC3I ratios of 1.09 ± 0.18 and 1.31 ± 0.16, respectively. Rn-C 0.1 µM and NpQ 10 µM did not prevent autophagy induction with fold changes in LC3II/LC3I ratios of 2.63 ± 0.38 and 2.61 ± 0.28 respectively (Figure 8B).

### 3.5. LC3 Localization with P62

The co-localization coefficient (CC) for LC3 and P62 represented in arbitrary units (AU) for the control HT-22 cells was 0.05 ± 0.01 (AU) (Figure 9). In contrast, treatment with glutamate showed a CC of 0.15 ± 0.03 (AU), which was statistically significant with ANOVA using Dunnett’s post hoc test (*p* < 0.05).

## 4. Discussion

We isolated and purified two naphthoquinones from Rn, Rn-C, and Rn-D and compared their activities to NpQ. We have shown for the first time that Rn-C at concentrations of 0.1–10 µM has neuroprotective effects against glutamate-induced ROS damage and toxicity in HT-22 cells, whereas Rn-D did not protect against glutamate toxicity (within the concentrations tested 0.1–10 µM). We also showed that NpQ had protective activity at the highest concentration tested 10 µM. 

The neuroprotective effect seen with Rn-C mirrors what was previously seen with the crude extract of Rn [16,17], preventing glutamate-induced toxicity and reducing ROS-induced stress in HT-22 cells; furthermore, other studies with Rn-C have seen similar protective effects in different cell models of neurodegeneration [18] and animal models [19]. A recent study in rats had shown Rn-C to have anti-parkinsonian effects, while not affecting liver enzymes or other organs; the study also showed that Rn-C was able to increase the levels of monoamines in the brain [37]. 

The molecular mechanisms underlying the death of HT-22 cells subjected to glutamate have been debated, mainly because the features of cell death manifest as both apoptotic and non-apoptotic [38,39,40]; furthermore, treatment with caspase inhibitors has provided mixed results, with some studies finding them to be protective [41,42]. In contrast, some others find them ineffective. Previous studies from this laboratory have reported caspase-12 cleavage, annexin V detection in glutamate treated HT-22 cells [21], and apoptosis-inducing factor (AIF) activation and caspase-independent cell death [43]. This all suggests that oxidative stress, caused by ER stress and mitochondrial leakage, induces a caspase-dependent or independent form of apoptosis; moreover, other conditions of cell death have been linked to glutamate toxicity in HT-22 cells, including the calpain, cathepsin, and the ubiquitin-proteasome system [44,45]. In this study, we have shown that glutamate induces HT-22 cell death via an apoptotic mechanism, shown by the Annexin V staining which is in line with other studies [46]; we also show that Rn-C is capable of dose-dependently preventing this apoptotic cell death. There is a small discrepancy between the MTT data and the flow cytometry data with regards to the toxicity of glutamate in HT-22 cells. This is likely due to the difference between measuring cell viability indirectly via MTT metabolism and measuring apoptosis using Annexin V and PI staining. In the flow cytometry study at both concentrations, 5 and 10 mM glutamate, there was cellular apoptosis measured and Rn-C was able to ameliorate the cell death/ apoptosis. This indicates that Rn-C is indeed cytoprotective against the ROS-inducing effects of glutamate and subsequent apoptotic cell death in HT-22 cells.

Another form of cell death other than necrotic and apoptotic resulting from mitochondrial and ER stress seen in HT-22 cells treated with glutamate is autophagic cell death; previous studies have shown that autophagy plays a significant component of the cell death seen in HT-22 cells subjected to glutamate treatment [12,21,40,47,48]. In this study, we show that autophagy is activated by glutamate treatment shown by LC3BI conversion to LC3BII and LC3B localization with P63. Furthermore, we showed that Rn-C (1 and 10 µM) could prevent LC3BI conversion to LC3BII, and Rn-C (10 µM) prevented LC3B localization with P63.

Autophagy can be both a mechanism of cell death and cell survival. In the case of prolonged ER stress, activation of autophagy can protect the cell by degrading the excess of misfolded proteins; however, in the case of ROS generation, autophagic cell death can result [49,50,51]. In this study, we see both ROS and autophagy induction in response to glutamate toxicity. In line with other studies, we also see ERK1/2 phosphorylation [12]; furthermore, we see that Rn-C can prevent ERK phosphorylation which has been seen previously in studies of Rn-C’s neuroprotective properties [18]. ERK1/2 phosphorylation is usually associated with cell survival, much like autophagy induction; however, ERK1/2 activation results in non-apoptotic neuronal cell death, as previously described [52]. Therefore, the inhibition of ROS production, ERK1/2 activation, and the inhibition of autophagy by Rn-C is cytoprotective. 

ER stress and its presence identified using CHOP as a marker has been shown in multiple studies of glutamate toxicity in HT-22 cells [20,30,53]. Many studies identify ER stress inhibition as a mechanism by which glutamate toxicity in HT-22 cells can be prevented [17,19,26,42]; however, in this study, we observed protection against cell death, but rather than seeing a reduction in CHOP expression as seen in previous studies, we identified a significant increase in CHOP expression, over and above that of glutamate, indicating that Rn-C is exacerbating ER stress, while still preventing cell death. We hypothesize that inhibition of autophagy could exacerbate ER stress, as the misfolded proteins were allowed to build up, resulting in increased CHOP expression [54]. At lower doses of Rn-C, we observed a reduction in CHOP expression with no autophagy inhibition, yet we still observed cell protection against glutamate.

Similarly, this was also the case with NpQ; therefore, the neuroprotection provided by Rn-C at either high or low doses is not, or at least only in part, a result of autophagic cell death inhibition, since we identified cytoprotection at the lower concentrations where there was no activation of autophagy. Rn-C could be more likely acting on other pathways related to ROS signaling to provide the protection identified in this study since ROS levels were reduced at both high and low concentrations of Rn-C; it is, therefore, possible that Rn-C is protecting against mitochondrial leakage, which would result in ROS and caspase-dependent cell death, as seen in other neuronal studies of Rn-C [18,19]. Further studies into Rn-C’s neuroprotective effects and resulting mechanisms will be required to ascertain whether it is a likely candidate molecule for further development as a neuroprotective drug.

## 5. Conclusions

In conclusion, we have identified Rn-C as an anti-ROS agent with cytoprotective effects against glutamate-induced toxicity/apoptosis in HT-22 cells. Rn-C inhibition of autophagy may be cytoprotective, as it coincides with the prevention of apoptosis. High concentrations of Rn-C may potentiate ER stress as seen by the CHOP expression by inhibiting autophagy; however, this does not appear to result in further cell death; thus, Rn-C is likely to protect the cells against ROS-related pathways. 

## Figures and Tables

**Figure 1 pharmaceuticals-15-00627-f001:**
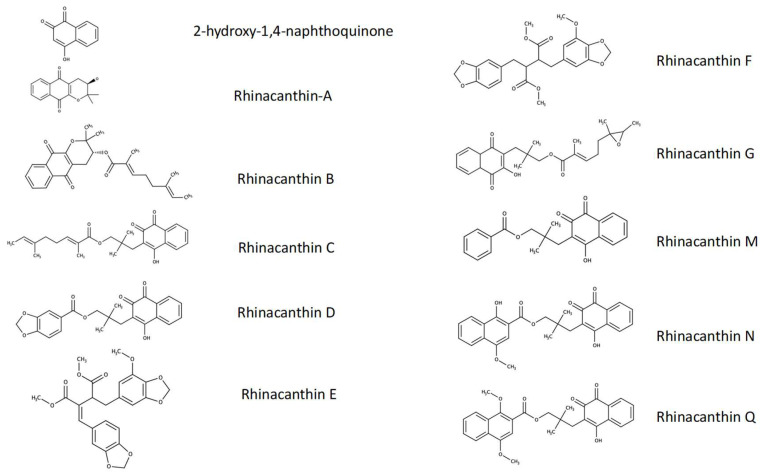
Structures of compounds that have been isolated from *R. nasutus*.

**Figure 2 pharmaceuticals-15-00627-f002:**
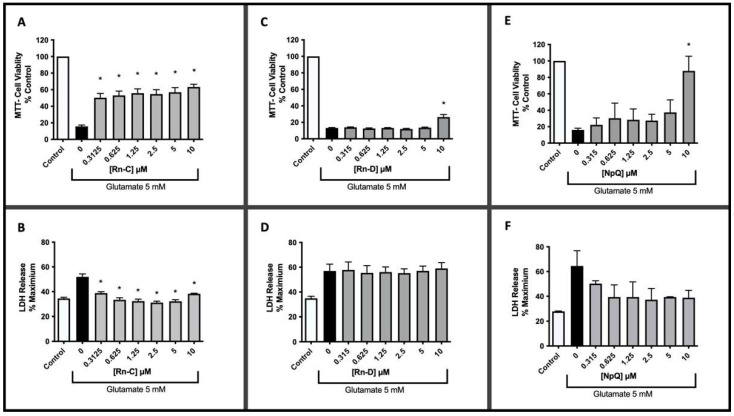
The protective effects of Rn-C, Rn-D, and NpQ (0–10 µM) against 5 mM glutamate toxicity in HT22 cells. (**A**) MTT assay of HT-22 cells with glutamate and Rn-C. (**B**) LDH assay of HT-22 cells with glutamate and Rn-C. (**C**) MTT assay of HT-22 cells glutamate and Rn-D. (**D**) LDH assay of HT-22 cells with glutamate and Rn-D. (**E**) MTT assay of HT-22 cells glutamate and NpQ. (**F**) LDH assay of HT-22 cells with glutamate and NpQ * (*p* < 0.05) ** (*p* < 0.01) *** (*p* < 0.001) **** (*p* < 0.0001) statistically significant change compared to cells treated with glutamate treated cells ANOVA followed by Dunnett’s post hoc test. *n* = 3.

**Figure 3 pharmaceuticals-15-00627-f003:**
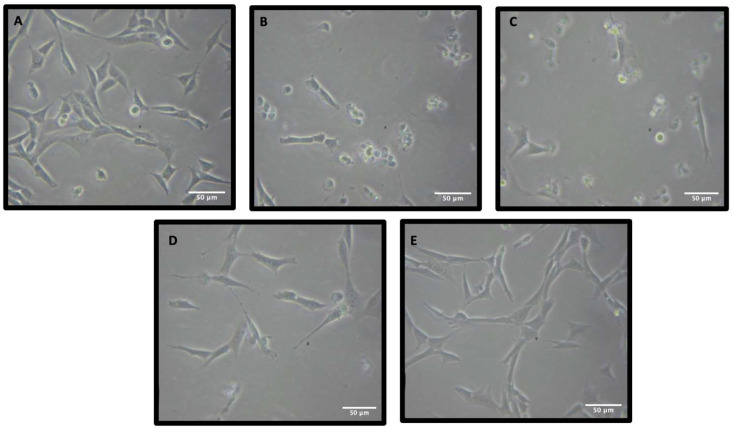
Micrographs showing the protective effect of Rn-C against 5 mM Glutamate. (**A**) Control HT-22 cells. (**B**) HT-22 cells were treated with 5 mM Glutamate for 18 h. (**C**) HT-22 cells treated with 5 mM. glutamate and 0.1 µM Rn-C for 18 h. (**D**) HT-22 cells were treated with 5 mM glutamate and 1 µM Rn-C for 18 h. (**E**) HT-22 cells were treated with 5 mM glutamate and 10 µM Rn-C for 24 h.

**Figure 4 pharmaceuticals-15-00627-f004:**
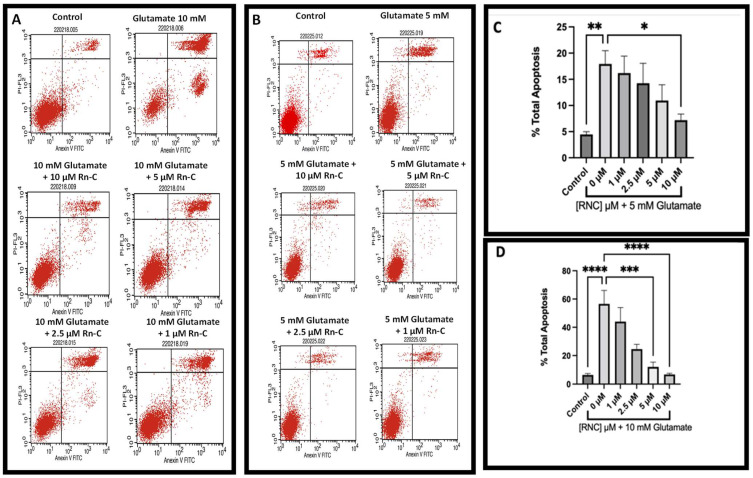
Flow cytometry showed the prevention of apoptosis (visualized by Annexin V and PI staining) resulting from glutamate toxicity in HT-22 cells by Rn-C between 10 and 1 µM. (**A**) Histograms of annexin V/PI-stained HT-22 cells treated with/without glutamate 10 mM and/or Rn-C (1–10 µM). (**B**) Histograms of annexin V/PI-stained HT-22 cells treated with/without glutamate 5 mM and/or Rn-C (1–10 µM). (**C**) Total apoptosis in HT-22 cells treated with 5 mM glutamate ± Rn-C (1–10 µM). (**D**) Total apoptosis in HT-22 cells treated with 5 mM glutamate ± Rn-C (1–10 µM). * (*p* < 0.05) ** (*p* < 0.01) *** (*p* < 0.001) **** (*p* < 0.0001) statistically significant change compared to cells treated with glutamate alone ANOVA followed by Dunnett’s post hoc test. *n* = 3.

**Figure 5 pharmaceuticals-15-00627-f005:**
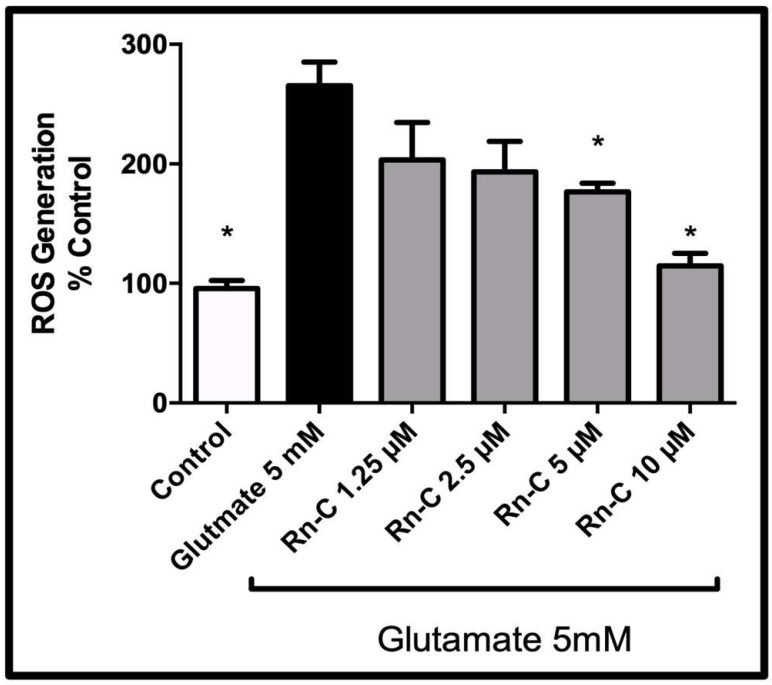
Rn-C induced reduction of ROS production caused by 5 mM glutamate in HT-22 cells. * Statistically significant reduction in ROS compared to 5 mM glutamate—ANOVA followed by Dunnett’s post hoc test *p* < 0.05 *n* = 5.

**Figure 6 pharmaceuticals-15-00627-f006:**
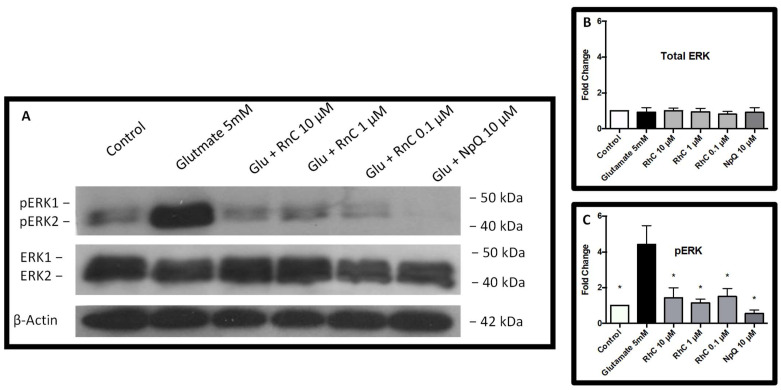
ERK1/2 activation by glutamate. (**A**) Western blotting showing the levels of ERK1/2 and pERK1/2 in HT-22 cells after treatment with Glutamate, or Glutamate and Rn-C (0.1–10 µM) or NpQ (10 µM). (**B**) Densitometry of total ERK1/2 expression in HT-22 cells. (**C**) Densitometry of pERK1/2 expression in HT-22 cells. * ANOVA with Dunnett’s post hoc test compared to 5 mM glutamate *p* value < 0.05 *n* = 5.

**Figure 7 pharmaceuticals-15-00627-f007:**
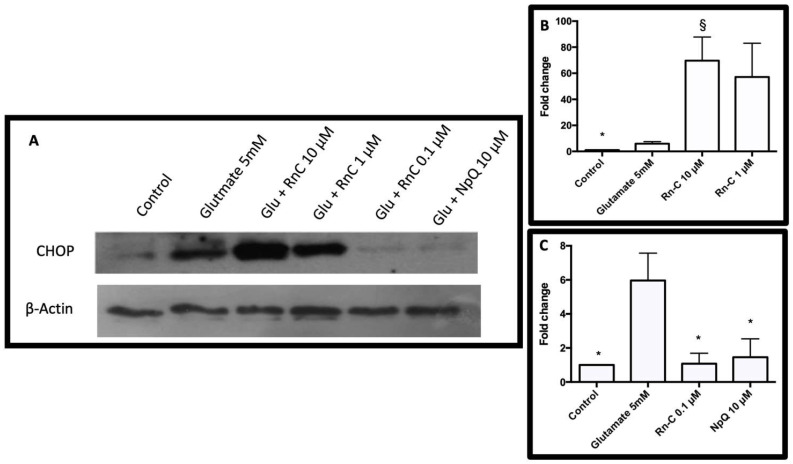
(**A**) Western blot of CHOP expression in HT-22 cells after treatment with glutamate. (**B**) Effect of Rn-C (10–1 µM) on CHOP Expression * Statistically Significant reduction compared to glutamate treated cells-ANOVA with Dunnett’s post hoc test *p* < 0.05 § Statistically Significant increase compared to glutamate treated cells-ANOVA with Dunnett’s post hoc test *p* < 0.05. (**C**) Effect of Rn-C 0.1 µM) and NpQ (10 µM) on CHOP Expression. * Statistically Significant reduction compared to glutamate treated cells-ANOVA with Dunnett’s post hoc test *p* < 0.05 (*n* = 3).

**Figure 8 pharmaceuticals-15-00627-f008:**
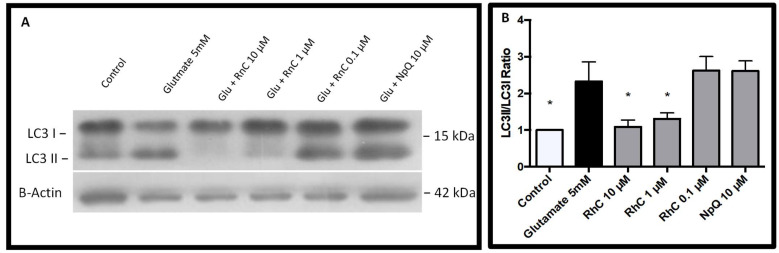
Autophagy induction by glutamate in HT-22 cells. (**A**) Western blot showing LC3-I and LC3-II expression in HT-22 cells, treated with glutamate 5 mM with and without RnC (10–0.1 µM) (**B**) Densitometry analysis showing LC3II/LC3I ratio * ANOVA analysis compared to 5 mM glutamate with Dunnett’s post hoc test *p* < 0.05 (*n* = 3).

**Figure 9 pharmaceuticals-15-00627-f009:**
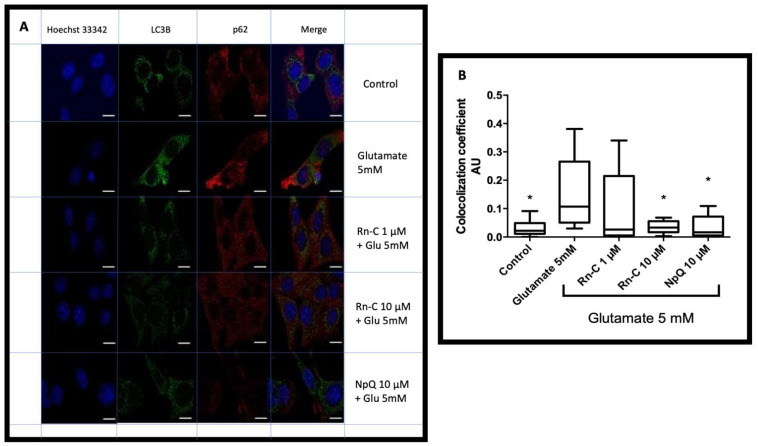
LC3 colocalization with p62. (**A**) Representative images of LC3B and p62 expression in HT- cells treated with glutamate and/or Rn-C (1–10 µM) or NpQ 10 µM Scale bar—10 µm. (**B**) Analysis of colocalization between LC3B and p62 * significant decrease in colocalization compared to glutamate treated HT-22 cells ANOVA with Dunnett’s post hoc test *p* < 0.05.

## Data Availability

All data sets are available with reasonable request to the corresponding authors.

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
