# Peer review of "Rhinacanthin-C but Not -D Extracted from Rhinacanthus nasutus (L.) Kurz Offers Neuroprotection via ERK, CHOP, and LC3B Pathways"

_pharmaceuticals, 2022, doi:10.3390/ph15050627_

Round 1
Reviewer 1 Report
Rakkhittawattana and coworkers report the neuroprotective properties and molecular mechanisms of rhinacanthin-C isolated from Rhinacanthus nasutus. Due the progressive aging of the worldwide population, the development of new neuroprotective drugs for prevention and treatment of neurodegenerative diseases is very relevant. The manuscript is well written and has a straightforward description of methods and findings, as well as an interesting discussion. I would definitely recommend it for publication after authors be addressing the following major and minor concerns.
Major concerns:
- Figure 2: Authors present all the characterization data from rhinacanthin-C and –D in a one single poor resolution figure. It is impossible to check peaks and values either on Figure2B, 2C, 2D, nor 2E. Authors must present all the characterization data as independent and high resolution figures. I suggest that the improved figures be presented as a supplementary material. Additionally, at line 126 authors mentioned that the acquisition of the 1H-NMR spectra were performed in a Bruker Advance NMR equipment at 500 MHz. Results of figure 2, though, are related to acquisitions made at 400 MHz.
- Authors should present the 13C-NMR characterization data/spectra for both target compounds.
- Please cite the studies that described – for the first time - the characterization data of both compounds on lines 257 and 260, respectively (doi: 10.1021/np9601871).
- Lines 484-488: Authors should support the following thoughts with references. “…We hypothesize that inhibition of autophagy could exacerbate ER stress, as the misfolded proteins were allowed to build up, resulting in increased CHOP expression. At lower doses of Rn-C, we observed a reduction in CHOP expression with no autophagy inhibition, yet we still observed cell protection against glutamate.”
Minor concerns:
- Figure 1 has a poor quality. Authors should improve the resolution of figure 1.
- Figure 1: The hydrogen of the hydroxyl groups from rhinacanthin C, -D, -G, and –M are missing. Structures have to be fixed.
- Line 77: “Rhinacanthus nasutus” should be “Rhinacanthus nasutus”.
- Figure 2: Please double check figures 2-5 captions. Lack of punctuations. Same for figure 8 (line 389), 9 (line 407), and 10 (line 419).
- Line 400: “(Figure 7B)” should be “(Figure 9B)”.
- Line 456: “amorylate” should be “ameliorate”.
Author Response
Reviewer 1:
Rakkhittawattana and coworkers report the neuroprotective properties and molecular mechanisms of rhinacanthin-C isolated from Rhinacanthus nasutus. Due the progressive aging of the worldwide population, the development of new neuroprotective drugs for prevention and treatment of neurodegenerative diseases is very relevant. The manuscript is well written and has a straightforward description of methods and findings, as well as an interesting discussion. I would definitely recommend it for publication after authors be addressing the following major and minor concerns.
Major concerns:
- Figure 2: Authors present all the characterization data from rhinacanthin-C and –D in a one single poor resolution figure. It is impossible to check peaks and values either on Figure2B, 2C, 2D, nor 2E. Authors must present all the characterization data as independent and high resolution figures. I suggest that the improved figures be presented as a supplementary material. Additionally, at line 126 authors mentioned that the acquisition of the 1H-NMR spectra were performed in a Bruker Advance NMR equipment at 500 MHz. Results of figure 2, though, are related to acquisitions made at 400 MHz.
Figure 2 was removed, and better figures have been added to the supplementary materials.
- Authors should present the 13C-NMR characterization data/spectra for both target compounds.
Rh-C is a known compound, with 3H and 13C- NMR already carried out (Sendl et al 1996 and Wu et al 1998). Therefore for the purpose of confirming its extraction, we feel 3H NMR + mass spec and HPLC analysis is more than enough to confirm we have extracted and purified this specific compound. Thus the addition of 13C- NMR does not add anything further to the manuscript.
- Please cite the studies that described – for the first time - the characterization data of both compounds on lines 257 and 260, respectively (doi: 10.1021/np9601871).
The citation was added
- Lines 484-488: Authors should support the following thoughts with references. “…We hypothesize that inhibition of autophagy could exacerbate ER stress, as the misfolded proteins were allowed to build up, resulting in increased CHOP expression. At lower doses of Rn-C, we observed a reduction in CHOP expression with no autophagy inhibition, yet we still observed cell protection against glutamate.”
A citation has been added
Minor concerns:
- Figure 1 has a poor quality. Authors should improve the resolution of figure 1.
We have added a higher resolution Figure
- Figure 1: The hydrogen of the hydroxyl groups from rhinacanthin C, -D, -G, and –M are missing. Structures have to be fixed.
The structures of all the compounds in figure 1 have been checked against pubchem records and are now correct
- Line 77: “Rhinacanthus nasutus” should be “Rhinacanthus nasutus”.
This has been corrected
- Figure 2: Please double check figures 2-5 captions. Lack of punctuations. Same for figure 8 (line 389), 9 (line 407), and 10 (line 419).
These have been checked and corrected
- Line 400: “(Figure 7B)” should be “(Figure 9B)”.
Thank you for spotting this mistake, it has now been corrected
- Line 456: “amorylate” should be “ameliorate”.
This has been amended.

Reviewer 2 Report
The authors presented a research article on Rhinacanthin-C but not -D extracted from Rhinacanthus nasutus (L.) Kurz offers neuroprotection via ERK, CHOP, and LC3B 3 pathways. The topic is interesting and well within the aims and scopes of the Journal.
The manuscript needs some changes and implementations before it can be considered for publication in this eminent Journal.
All the comments are highlighted through track changes in the manuscript, that should be addressed.

Author Response
Reviewer 2:
Title:
The name of the plant must be italic
This has been amended
Page 2 line 63:
The science does not believe on future observation. Revise the sentence as "Several natural products have been isolated from Rn, with some of the most abundant being the rhinacanthins" with proper citation
We are not sure what is meant by “The science does not believe on future observation”
Our statement implies that many rhinacanthin compounds have already been identified from this plant, not that they will be identified in the future.
We have added some further citations to clarify.
Page 6 line 260:
Add the 13C NMR and HRMS of the Rn-C and Rn-D here.
This is now in the supplementary data as suggested above
The Authors should provide HRMS spectra with at least four decimal digits, otherwise the structure cannot be confirmed.
This has been added in the supplementary data as suggested
This is experimental data of the compounds and better to shift to the experimental part.
This has been moved as suggested
Page 6 line 266:
Write the brief information about the identification of both compounds
This section was amended with changes asked for above, there is no just a brief section about the identification of the compounds, with all the main data in the supplementary figures.
Page 7 Figure 2:
As the compounds are known and already isolated from the same plant, therefore, it will be better to move the figure 2 to the supporting material.
This has been done as suggested
References:
The journal' names must be in abbreviation form and according to the style of the journal guidelines.
The references have been amended

Round 2
Reviewer 1 Report
I believe the manuscript is acceptable for publication as it is, even though this reviewer still believe that the insertion of 13C NMR characterization data would be highly recommended.
Author Response
Thank you for you comments on our manuscript. While we agree that 13C NMR is a useful tool. 13C data from this compound has already been published on several occasions, and the NMR + mass spec in this study were just to confirm we isolated the correct compound 13C NMR is a bit of over kill in our humble opinion.